# Survival of HT29 cancer cells is influenced by hepatocyte growth factor receptor inhibition through modulation of self-DNA-triggered TLR9-dependent autophagy response

**Bettina Bohusné Barta**[1], **Ágnes Simon**[1], **Lőrinc Nagy**[1], **Titanilla Dankó**[2], **Regina Eszter Raffay**[2], **Gábor Petővári**[2], **Viktória Zsiros**[3], **Anna Sebestyén**[2], **Ferenc Sipos**[1]*, **Györgyi Műzes**[1]

1 Department of Internal Medicine and Hematology, Semmelweis University, Budapest, Hungary, 2 1st Department of Pathology and Experimental Cancer Research, Semmelweis University, Budapest, Hungary, 3 Department of Anatomy, Histology and Embryology, Semmelweis University, Budapest, Hungary

☯ These authors contributed equally to this work.

* dr.siposf@gmail.com

**Data Availability Statement:** All relevant data are within the paper.

## Abstract

HGFR activation drives the malignant progression of colorectal cancer, and its inhibition displays anti-autophagic activity. The interrelated role of HGFR inhibition and TLR9/autophagy signaling in HT29 cancer cells subjected to modified self-DNA treatments has not been clarified. We analyzed this complex interplay with cell metabolism and proliferation measurements, TLR9, HGFR and autophagy inhibitory assays and WES Simple Western blot-based autophagy flux measurements, gene expression analyses, immunocytochemistry, and transmission electron microscopy. The overexpression of MyD88 and caspase-3 was associated with enhanced HT29 cell proliferation, suggesting that incubation with self-DNAs could suppress the apoptosis-induced compensatory cell proliferation. HGFR inhibition blocked the proliferation-reducing effect of genomic and hypermethylated, but not that of fragmented DNA. Lowest cell proliferation was achieved with the concomitant use of genomic DNA, HGFR inhibitor, and chloroquine, when the proliferation stimulating effect of STAT3 overexpression could be outweighed by the inhibitory effect of LC3B, indicating the putative involvement of HGFR-mTOR-ULK1 molecular cascade in HGFR inhibitor-mediated autophagy. The most intense cell proliferation was caused by the co-administration of hypermethylated DNA, TLR9 and HGFR inhibitors, when decreased expression of both canonical and non-canonical HGFR signaling pathways and autophagy-related genes was present. The observed ultrastructural changes also support the context-dependent role of HGFR inhibition and autophagy on cell survival and proliferation. Further investigation of the influence of the studied signaling pathways and cellular processes can provide a basis for novel, individualized anti-cancer therapies.

**Funding:** The study was funded by the StartUp Program of Semmelweis University Faculty of Medicine (CO No.: 11720, Ikt.sz.: 5127/AOKGIE/2018; SE10332470) awarded to FS and GM. The funders had no role in study design, data collection and analysis, decision to publish, or preparation of the manuscript.

**Competing interests:** The authors have declared that no competing interests exist.

## Introduction

The *c-Met* proto-oncogene encodes a transmembrane receptor tyrosine-kinase (RTK) protein (C-MET; HGFR: hepatocyte growth factor receptor) containing two disulphide linked sub-units (alpha and beta). In physiological circumstances HGFR is usually expressed by epithelial, muscle, hematopoietic, immune, and nerve cells, among others. Regarding tumorigenesis, several tumor cells and the cellular components of the tumor-stroma also express HGFR [1]. Hepatocyte growth factor (HGF), the ligand of HGFR, is a pleiotropic cytokine mainly produced by mesenchymal cells, including fibroblasts and macrophages. HGF promotes several cellular functions, including survival, tissue protection, regeneration, and exerts anti-inflammatory activities [2]. Moreover, HGF regulates various immune functions, like cytokine production, cellular migration, and adhesion [3].

The activation of HGFR can materialize by the canonical (i.e. HGF binding to HGFR resulting in HGFR homodimerization) and non-canonical (i.e, HGFR dimerizes with different receptors) pathways [4]. HGFR activation drives the malignant progression of colorectal cancer (CRC) by promoting signaling cascades that mainly affect the survival, proliferation, motility, migration, and invasion of cancer cells [5]. Signaling within and beyond this pathway seems to be an important factor regarding systemic spread of metastases through induction of epithelial-to-mesenchymal transition [2]. HGFR inhibition was reported to sensitize HT29 colorectal cancer cells to irradiation by enhancing the formation of deoxyribonucleic acid (DNA) double strand breaks and possibly alleviating tumor hypoxia [6]. The pro-tumorigenic effects of the HGF/HGFR-system can be mediated by transcriptional activation, gene amplification, gene mutation or autocrine/paracrine HGF stimulation [7]. Aberrant HGF/HGFR activation has been observed in many solid tumor types (including hepatocellular, pancreatic ductal, and colorectal cancers), and promotes cellular proliferation and metastasis via growth factor and other oncogenic receptors [8]. Thus, HGF/HGFR inhibition has come up as targeted anticancer therapy.

The HGF/HGFR-system inhibitors can be classified as adenosine-triphosphate (ATP)-competitive and ATP non-competitive small molecule C-MET inhibitors, anti-HGF-, and anti-HGFR antibodies [9, 10]. Cross-talk between HGFR and epidermal growth factor receptor (EGFR) is also implicated in carcinogenesis [11]. HGFR amplification in metastatic CRC has been found to be an acquired response to EGFR inhibition, not a *de novo* phenomenon [12].

Autophagy is an evolutionarily conserved proteolytic process including lysosomal degradation and recycling impaired cellular components and energy to maintain homeostasis [13, 14]. Protective autophagy blockade has been applied simultaneously with either chemotherapies or targeted therapies to optimize their efficacy in different cancers in preclinical studies [15]. HGFR inhibition was shown to display different (i.e. inhibitory or activating) effects on autophagy in cancer cells [16, 17]. Recent findings also indicate that the HGFR/mechanistic target of rapamycin (mTOR)/Unc-51 Like Autophagy Activating Kinase 1 (ULK1) cascade is responsible for HGFR-mediated autophagy, hence targeting autophagy may potentiate antitumor activity of HGFR-tyrosine kinases against *Met*-amplified cancer cells [12, 15, 18].

Recently, longitudinal DNA methylation changes at HGFR has been shown to alter HGF/HGFR signaling cascade [19]. Furthermore, DNA aptamers have emerged as advantageous chemical substances for designing growth factor mimetics, including the ones for HGFR [20, 21]. The construction of effective inhibitors for HGF is an important issue in antitumor treatment. Generation of inhibitory DNA aptamers against human HGF would be useful as therapeutic agents for cancers [22].

In HT29 colon cancer cells, a close interplay between self-DNA-induced TLR9-signaling and autophagy response was found with notable effects on cell survival and differentiation

[23]. However, the interrelated role of HGFR inhibition and TLR9/autophagy signaling in HT29 colon cancer cells has not yet been clarified. Therefore, we aimed to assess this complex interaction in HT29 cells. Here we found evidence for a close interplay between the inhibition of HGFR canonical and non-canonical downstream signaling pathways and TLR9/autophagy response with remarkable influences on survival, metabolic activity, and proliferation of HT29 colon carcinoma cells subjected to intact or modified self-DNA treatments.

## Materials and methods

### Selection and maintenance of HT29 cell culture; self-DNA isolation

The selection of HT29 cells was made taking into account several aspects. There is basal TLR9 expression in HT29 cells, which is essential for induction with self-DNA [24]. Moreover, the MyD88-dependent and MyD88-independent TLR signaling pathways are intact in HT29 cells [25]. In HT29 cells, HGFR expression is high as compared to other CRC cell lines [26], and TLR and autophagy-mediated HGFR cross-activation is also present [27–29]. HT29 cells adequately represents sporadic colon cancers [30]. Not all colorectal cancer cell lines meet these criteria.

HT29 undifferentiated colon adenocarcinoma cell line was purchased from the 1st Department of Pathology and Experimental Cancer Research (Semmelweis University, Budapest, Hungary). The cells were maintained in RPMI 1640 medium (Sigma-Aldrich, USA) supplemented with 10% (v/v) fetal bovine serum (FBS; Standard Quality; PAA Laboratories GmbH, Austria), 125 μg/ml amphotericin B (Sigma-Aldrich, USA), and 160 μg/ml gentamycin (Sandoz, Sandoz GmbH, Austria). The medium was replaced every second day.

Genomic DNA was isolated from $5 \times 10^7$ steady state, proliferating HT29 cells. DNA isolation was performed by using High Pure PCR template preparation kit containing proteinase K (Roche GmbH, Germany). The DNA samples were treated with 5 μl RNase A/T1 Mix (Thermo Scientific, Germany). DNA concentration was determined by Nanodrop (Thermo Scientific, Germany).

### Fragmentation and hypermethylation of self-DNA for HT29 cell incubation

Genomic DNA was divided into three equal shares: the first one was neither fragmented nor hypermethylated (genomic DNA: gDNA). The second one was fragmented (fragmented-DNA: fDNA) by ultrasonic fragmentation for 2 min. The third share was hypermethylated (methylated-DNA: mDNA) using CpG methyltransferase M.SssI (New England Biolabs Ipswich, USA). Length of the fragmented DNA shares was determined by agarose gel electrophoresis. According to MALDI-TOF mass spectrometry measurements, the DNA samples were free of RNA, protein, or lipopolysaccharide contamination.

### HT29 cell treatments

To incubate with the DNA samples, $0.5 \times 10^6$ HT29 cells were seeded in a 12-well plate with RPMI 1640 supplemented with amphotericin B, gentamycin, and FBS, as previously described. After 24 hours, the medium was changed to RPMI 1640 supplemented with gentamycin but lacking FBS. Separate aliquots of 15 μg modified self-DNA were dissolved in 200 μl sterile phosphate buffered saline (PBS).

At 37˚C, HT29 cells were incubated with the various DNA samples in a humidified atmosphere containing 5% $CO_2$ and 95% $O_2$. Only 200 ul sterile PBS was added to the control cells.

Cells were washed twice with 5 ml sterile PBS and resuspended in a final volume of 5 ml PBS after 72 hours.

## Inhibition of TLR9- and HGFR-signaling

For inhibition of TLR9-, or HGFR-signaling, HT29 cells were pretreated with TLR9 antagonist (5 μM ODN2088; Invivogen, CA, USA), or 4,4'Diisothiocyanatostilbene-2,2'-disulfonic acid (DISU; 4 μM; D3514 Sigma-Aldrich, Budapest, Hungary; diluted in dimethyl sulfoxide /DMSO; Sigma-Aldrich Budapest, Hungary/) for 1 hour before treatments with DNAs. All treatments were performed in triplicate. Between plates, 2–2 samples received the same treatment to avoid possible manual errors in the treatments between plates.

## Autophagy inhibition and assessment of autophagic flux with WES Simple Western blot

Chloroquine, an anti-inflammatory substance is the most commonly used drug to asses autophagic flux because of its suitability in vivo. HT29 cells were started to be treated with chloroquine (10 μM; C6628 Sigma-Aldrich, Budapest, Hungary; diluted in DMSO) for 1 hour before treatments with DNA. All treatments were performed in triplicate. Between plates, 2–2 samples received the same treatment to avoid possible manual errors in the treatments between plates.

WES Simple (ProteinSimple 004–600, Minneapolis, MN, USA) analysis was also performed. A 12–230 kDa Separation Module (ProteinSimple SM-W004) was used for all the proteins (Anti-SQSTM1/p62 antibody [2C11]—BSA and Azide free /Abcam; ab56416/; LC3B (D11) XP Rabbit mAb /CellSignaling; #3868/; Anti-β-Actin (AC-74) Mouse mAb /SigmaAldrich; A2228/; GAPDH (14C10) Rabbit mAb /CellSignaling; #2118/), and either the Anti-Rabbit Detection Kit (ProteinSimple DM-001) or Anti-Mouse Detection Kit (ProteinSimple DM-002) were used, depending on the primary antibodies. Briefly, based on the used primary antibodies, 0.2 or 1 μg/μL cell lysates were diluted in 0.1× WES Sample Buffer (ProteinSimple 042–195), and Fluorescent Master Mix (1:4, ProteinSimple PS-FL01-8) was also added. Following a 5-minute incubation at 95˚C, the Antibody Diluent (ProteinSimple 042–203), primary and secondary antibodies, and chemiluminescent substrate were applied to the WES capillary plate. The WES system settings were (a) stacking and separation (395 V, 30 min.), (b) blocking (5 min.), (c) incubations with primary and secondary antibodies (30 min.) and (d) luminol/peroxide chemiluminescence detection (15 min.) (the exposure time was 2 sec.). The electropherograms were manually corrected if required for the evaluations. The treatment plan for HT29 cells is shown in **Table 1**.

## Cell viability and proliferation measurements

The use of the Alamar Blue assay served a dual purpose: partly to examine cell viability (metabolic activity) and partly to study cell proliferation.

The anti-proliferative effects of the 72h long treatments were measured after a 4h incubation period using Alamar Blue (Thermo Fisher Scientific, Budapest, Hungary). The fluorescence was measured at 570–590 nm (Fluoroskan Ascent FL fluorometer; Labsystems International Ltd., Budapest, Hungary) and the results were analyzed by Ascent Software.

As metabolic activity is not necessarily proportional to proliferative activity, direct cell counts (average cell numbers) were also performed in the examined cell groups to determine the proliferative activity compared to the control sample.

**Table 1. Treatment plan for HT29 cancer cells.** g/f/mDNA: genomic/fragmented/hypermethylated deoxyribonucleic acid; ODN: CpG oligonucleotide; DISU: 4,4'Diisothiocyanatostilbene-2,2'-disulfonic acid.

| Sample groups | gDNA | fDNA | mDNA | ODN2088 | DISU | Chloroquine |
|---|---|---|---|---|---|---|
| 1 | - | - | - | - | - | - |
| 2 | - | - | - | + | - | - |
| 3 | - | - | - | - | + | - |
| 4 | - | - | - | - | - | + |
| 5 | + | - | - | - | - | - |
| 6 | - | + | - | - | - | - |
| 7 | - | - | + | - | - | - |
| 8 | + | - | - | + | - | - |
| 9 | + | - | - | - | + | - |
| 10 | + | - | - | - | - | + |
| 11 | + | - | - | + | + | - |
| 12 | + | - | - | - | + | + |
| 13 | - | + | - | + | - | - |
| 14 | - | + | - | - | + | - |
| 15 | - | + | - | - | - | + |
| 16 | - | + | - | + | + | - |
| 17 | - | + | - | - | + | + |
| 18 | - | - | + | + | - | - |
| 19 | - | - | + | - | + | - |
| 20 | - | - | + | - | - | + |
| 21 | - | - | + | + | + | - |
| 22 | - | - | + | - | + | + |

## Total mRNA isolation and Nanostring analysis

Total mRNA from HT29 cells was extracted with RNeasy Mini Kit (Qiagen, USA) according to the prescription of the manufacturer. Quantitative (Nanodrop) and qualitative analysis (Bioanalyzer Pico 600 chip kit RNA program; RIN >8 in all cases) were performed.

mRNA samples required for gene expression assays of HT29 cells were prepared by tripling the treated groups. In HT29 samples, cell numbers ranged from 100000 to 11135000 per well, and the recovered mRNA concentration ranged from 8 to 256 ng / μl /sample. mRNAs recovered from triplicates were pooled and used in the Nanostring assay.

The custom mRNA Assay Evaluation panel (NA-SPRINT-CAR-1.0, nCounter SPRINT Cartridge) containing our custom gene code set (NA-XT-GXA-P1CS-04 nCounter GX Custom CodeSet) was designed by Nanostring (The order was placed through Biomedica Hungaria Ltd., Budapest, Hungary). The Nanostring experiments were carried out by RT-Europe Research Center Ltd. (Mosonmagyaróvár, Hungary; website: http://rt-europe.org/) as part of a contract work.

The criterion for selecting the genes to be examined was to establish an association between C-Met/HGFR and TLR9 signaling, apoptosis, cell proliferation, and autophagy.

The gene set contained the following genes (with probe NSIDs):

TLR9-signaling and NF-κβ activation: TLR9 (Toll-like receptor 9; NM_017442.2:985), MyD88 (Myeloid differentiation factor 88; NM_002468.3:2145), IRAK2 (Interleukin 1 receptor associated kinase 2; NM_001570.3:1285), TRAF6 (Tumor necrosis factor receptor associated factor 6; NM_145803.2:745), IL1β (Interleukin 1β; NM_000576.2:840), IL8 (Interleukin 8; NM_000584.2:25), NFkB (Nuclear factor-kB; NM_003998.2:1675).

Extrinsic and intrinsic apoptosis-related genes: CD95 (Fas; NM_152876.1:1740), CD95L (Fas-ligand; NM_000639.1:625), Cytochrom-c (NM_001916.4:344), Caspase-3 (NM_004346.3:2156).

Anti-apoptotic and autophagy suppressor genes: PI3KCA (Phosphoinositide 3-kinase; NM_006218.2:2445), Akt (Ak strain transforming; NM_001014432.1:1275), mTOR (Mechanistic/mammalian target of rapamycin; NM_004958.3:1865), Bcl-2 (B-cell lymphoma 2; NM_000657.2:5).

Pro-apoptotic and autophagy activator genes: MAPK (Mitogen-activated protein kinase; NM_002755.2:970), AMPK (AMP-activated protein kinase; NM_006251.5:366), Bax (BCL2 associated X; NM_138761.3:342).

Autophagy genes: Beclin1 (NM_003766.2:810), ATG16L1 (Autophagy related 16 like 1; NM_017974.3:2405), MAP1LC3B (Microtubule-associated proteins 1A/1B light chain 3B; NM_022818.4:1685), ULK1 (Unc-51 like autophagy activating kinase; NM_003565.1:465).

C-Met/HGFR and C-Met canonical and non-canonical signaling pathways: HGFR (NM_001127500.1:1925), PI3KCA (see above), STAT3 (Signal transducer and activator of transcription 3; NM_003150.3:2060), CD95 (see above).

Housekeeping genes: C1orf43 (NM_015449.2:477), CHMP2A (NM_014453.3:241), PSMB2 (NM_002794.3:639), RAB7A (NM_004637.5:277), REEP5 (NM_005669.4:280), SNRPD3 (NM_004175.3:309), VCP (NM_007126.2:615), VPS29 (NM_016226.4:565).

## Taqman real-time polymerase chain reaction analysis

For validating the NanoString gene expression analysis method, mTOR (ID: Hs00234508_m1), ATG16L1 (ID: Hs01003142_m1), LC3B (ID: Hs00797944_s1), BCN1 (ID: Hs01007018_m1), HGFR (ID: Hs01565584_m1), PI3KCA (ID: Hs00907957_m1), STAT3 (ID: Hs00374280_m1), CD95 (ID: 4331182 Hs00236330_m1), and TLR9 (ID: Hs00370913_s1) triplicated Taqman real-time polymerase chain reactions were used in an Applied Biosystems Micro Fluidic Card System. The measurements were performed using an ABI PRISM 7900HT Sequence Detection System as described in the product's User Guide (http://www.appliedbiosystems.com, California; United States). Gene expression levels for each individual sample were normalized to GAPDH (ID: Hs02786624_g1) expression. Mean relative gene expression was determined and differences were calculated using the 2-ΔC(t) method. The whole cycle number was 45.

## C-MET, TLR9 and autophagy immunocytochemistry

To detect HGFR, TLR9, and autophagy-associated ATG16L2, Beclin1, and LC3 protein expression, HT29 cell smears were incubated with rabbit polyclonal anti-Met culture supernatant antibody (1:100, Clone: C-12, Santa Cruz Biotechnology Inc.), mouse anti-human monoclonal anti-TLR9 antibody (20 μg/mL; LS-B2341, clone: 26C593.2; LifeSpan BioSciences, USA), and anti-ATG16L1-, anti-BECN1-, and anti-MAP1LC3B antibodies (1:200, Antibody Verify, LA, USA) at 37°C for 1 hour. After three rounds of PBS rinsing, cell smears were treated for 40 minutes with an anti-rabbit EnVision polymerHRP conjugate kit (K4003, DAKO). Secondary immunodetection was performed according to the manufacturer's instructions using EnVision System Labeled Polymer–HRP K4001 (Anti-Mouse 1/1; DAKO). A Liquid DAB+ Substrate Chromogen System was used to convert the signal (DAKO). Hematoxylin co-staining was performed following rinsing in PBS. Smears of cells were then digitalized and analyzed using the CaseViewer software on a high-resolution PANNORAMIC 1000 FLASH DX instrument (3DHISTECH Ltd., Budapest, Hungary) (3DHISTECH Ltd., Budapest,

Hungary). Immunocytochemistry analyses were performed under contract (Pathology Laboratory, Heim Pál National Institute of Pediatrics, Budapest, Hungary).

## Cell counting and interpretation of immunoreactions

At 200x magnification, 10 fields of view and at least 100 cells (mainly 110 cells) per field of view were examined in a semiquantitative manner in each digitalized sample. The percentage of immunopositivity and non-immunoreactive HT29 cells was determined.

In the case of the TLR9 and HGFR immune response, weak, moderate and strong membrane staining and perinuclear cytoplasm staining were examined. As for autophagy, weak, moderate and strong ATG16L1 and Beclin1 homogenous or spotted immunoreactions were detected in the cytoplasm. In case of LC3, weak, moderate and strong punctuated or spotted cytoplasmic immunoreactions were observed.

The notation "- / +" indicates non-immunoreactive and weakly immunopositive cells. The designation "++ / +++" indicates moderately or strongly immunopositive cells.

## Transmission electron microscopy for evaluation of autophagy

For 60 minutes, HT29 cells in the wells were fixed in 2% glutaraldehyde (0.1M Millonig buffer, pH 7.4). Following three 5-minute washes with 0.1 M phosphate buffer and 0.1 M pH 7.2 sodium cacodylate buffer, the samples were post-fixed for 60 minutes at 4˚C in the dark with 1% osmium tetroxide in 0.1 M sodium-cacodylate buffer. After three 5-minute washes with sodium-cacodylate buffer (pH 7.4), cells were centrifuged and embedded in 10% gelatin in phosphate buffer (pH 7.4). Following dehydration in progressively increasing concentrations of alcohol, the samples were embedded in Poly/Bed epoxy resin. Contrast staining of ultrathin sections (70–80 nm) with uranyl acetate and lead citrate, respectively. JEM-1200EXII Transmission Electron Microscope was used to conduct ultrastructural analyses (JEOL, Akishima, Tokyo, Japan).

In five HT29 cells per sample, the average number of autophagic vacuoles was counted (mean ± SD/cell).

## Semithin sections

From the HT29 cell blocks fixed for TEM semithin sections were cut for viewing by digital microscope. The sections were stained with toluidine blue (toluidine blue O 4 g, pyronin 1 g, borax 5 g in distilled water). Semithin sections were then digitalized using high-resolution PANNORAMIC 1000 FLASH DX instrument (3DHISTECH Ltd., Budapest, Hungary), and analyzed with CaseViewer software (3DHISTECH Ltd., Budapest, Hungary).

## Statistical analysis

At least three independent experiments were conducted. Cell viability, cell number, and proliferation were expressed as means ± SD, while immunocytochemical results were measured semiquantitatively. Chi2-test, Student's t-test and one-way ANOVA with Tukey HSD test were used for statistical analyses. $P < 0.05$ was considered as statistically significant. Regrading NanoString gene expression analysis, after importing RCC files to the nSolver Analysis Software, quality checking was performed. Then agglomerative cluster heat maps were created. Euclidean distance metric was used to calculate the distance between two samples (or genes) as the square root of the sum of squared differences in their log count values. Average linkage method was used to calculate the distance between two clusters. In case of WES Simple

Western blot, the area of the tested proteins was multiplied by the values of the β-actin area for graphical representation.

## Ethics

This article does not contain any studies with human or animal subjects. In accordance with the standard operating procedure of the Institutional Review Board, the submission of the manuscript was approved.

## Results

### Cell viability and proliferation measurements

gDNA alone, and combined with ODN2088, DISU, or chloroquine treatment groups increased the metabolic activity of HT29 cells, respectively. However, when TLR9 or autophagy inhibitor treatments combined with gDNA were also combined with DISU, cell viability was significantly reduced.

In contrast to metabolic activity, gDNA administration reduced the proliferation of HT29 cells. After co-administration of ODN2088 and DISU, the inhibitory effect of gDNA on cell proliferation was significantly reduced. ODN2088 or DISU treatments in combination with gDNA separately decreased the inhibitory effect of gDNA on cell proliferation, but when used together they were much more effective. When gDNA, DISU, and chloroquine treatments were co-administered, the most effective inhibition of HT29 cell proliferation with high metabolic activity was observed.

fDNA treatment alone slightly increased cell viability, but when used in combination with TLR9 inhibitor metabolic activity was significantly increased, and moderately increased with DISU or chloroquine. In case of fDNA/ODN2088 and fDNA/chloroquine combinations, however, as a result of DISU, the metabolic activity of HT29 cells dropped to the level of fDNA control samples.

In fDNA control samples, HT29 cell proliferation increased slightly, but decreased after combination with either treatment, but to a different extent. In the case of fDNA/ODN2088 combination, the decrease in cell proliferation did not change significantly after DISU administration. However, in the case of fDNA/chloroquine combination, DISU reduced HT29 cell proliferation less.

Of all DNA types, mDNA increased the metabolic activity of cells the most. During the treatments, the degree of metabolic activity remained unchanged (DISU, ODN2088/DISU) or increased (ODN2088, chloroquine) as compared to the mDNA control, and showed a significant decrease only when DISU and chloroquine was co-administered.

mDNA treatment slightly reduced cell proliferation, and a decrease was observed with all treatments, most notably with inhibition of autophagy. However, a significant increase in cell proliferation was observed with mDNA/DISU or mDNA/ODN2088/DISU treatments. In the case of mDNA/ODN/DISU co-administration, interestingly, the increase of cell proliferation occurred with a significant decrease in metabolic activity. Viability, cell number, and proliferation data are illustrated in **Table 2** and **S1 Fig**. The one-way ANOVA and Tukey HSD test results can be seen in **S1 Table**.

### Nanostring and Taqman gene expression analyses

Regarding TLR9 mRNA expression, g-, f- and mDNA treatments resulted in TLR9 upregulation as compared to untreated control cells (**Fig 1A**). As for HGFR gene expression, gDNA treatment did not increase the expression as compared to untreated control, whereas fDNA

**Table 2. Numerical data of viability, cell number, and proliferation.** *represents significant alteration as compared to K (control), non-treated sample (p‹0.05). g/f/mDNA: genomic/fragmented/hypermethylated deoxyribonucleic acid; ODN: CpG oligonucleotide; DISU: 4,4'Diisothiocyanatostilbene-2,2'-disulfonic acid; C: chloroquine; SD: standard deviation.

| Sample | Alamar Blue (mean% ± SD) | Average cell number / 350 µl (±SD) | Proliferation% (± SD) |
|---|---|---|---|
| K | 100 ± 1.1 | 800000 ± 8800 | 100 ± 1.1 |
| O | 120.17 ± 4.5* | 760000 ± 32680 | 95 ± 4.3 |
| D | 111.41 ± 3.8 | 810000 ± 25920 | 101.25 ± 3.2 |
| C | 116.23 ± 2.9* | 775000 ± 31775 | 96.87 ± 4.1 |
| Kg | 127.51 ± 3.1* | 220000 ± 9900 | 27.5 ± 4.5* |
| Kf | 112.61 ± 2.2 | 855000 ± 31635 | 106.87 ± 3.7 |
| Km | 147.87 ± 3.4* | 720000 ± 19440 | 90 ± 2.7* |
| gO | 139 ± 3.1* | 270000 ± 3780 | 33.75 ± 1.4* |
| gD | 134.44 ± 2.7* | 310000 ± 6510 | 38.75 ± 2.1* |
| gC | 123.55 ± 3.1* | 230000 ± 4370 | 28.75 ± 1.9* |
| gOD | 75.75 ± 2.6* | 690000 ± 24840 | 86.25 ± 3.6* |
| gDC | 90.99 ± 3.3 | 100000 ± 1600 | 12.5 ± 1.6* |
| fO | 198,02 ± 4.7* | 745000 ± 23840 | 93.12 ± 3.2 |
| fD | 120.87 ± 3.7* | 665000 ± 21945 | 83.12 ± 3.3* |
| fC | 121.18 ± 2.5* | 560000 ± 13440 | 70 ± 2.4* |
| fOD | 99.61 ± 2.3 | 730000 ± 26280 | 91.25 ± 3.6 |
| fDC | 107.62 ± 3.2 | 640000 ± 17920 | 80 ± 2.8* |
| mO | 155.15 ± 4.1* | 740000 ± 25160 | 92.5 ± 3.4 |
| mD | 141.85 ± 3.9* | 875000 ± 36750 | 109.37 ± 4.2 |
| mC | 183.48 ± 4.6* | 560000 ± 16240 | 70 ± 2.9* |
| mOD | 90.12 ± 2.5 | 1140000 ± 60420 | 142.5 ± 5.3 * |
| mDC | 92.34 ± 3.1 | 580000 ± 9860 | 72.5 ± 1.7* |

and mDNA treatments upregulated the HGFR gene expression. Otherwise, incubation with gDNA displayed similar gene expression profile as seen in control samples (**Fig 1A**). In case of fDNA treatment, all observed genes displayed upregulated mRNA expressions except that of IL1β. Extrinsic and intrinsic apoptosis-related genes (i.e., Bcl2, CD95, caspase-3) were strongly upregulated. Autophagy-related (ULK1), TLR9-signaling (TRAF6), C-MET-signaling/anti-apoptotic (PI3K and HGFR), and apoptosis-related (CD95L) genes showed moderately strong upregulation, while autophagy-related (ATG16L1, MAP1LC3B, Beclin1), TLR9-signaling (MyD88), proapoptotic (AMPK), and C-MET-signaling-related (STAT3) genes displayed only weak upregulation (**Fig 1A**). In case of incubation with mDNA, anti-apoptotic Bcl2 gene showed strong overexpression, while TLR9-signaling (IL8, MyD88), anti-apoptotic (Akt), pro-apoptotic (MAPK), C-MET-signaling (HGFR), and autophagy-related (MAP1LC3B) genes showed moderate overexpression (**Fig 1A**).

Regarding the effect of combined HGFR inhibition and modified DNA treatments on C-MET canonical and non-canonical signaling, co-administration of DISU and gDNA was found to increase STAT3 and CD95, slightly increase PI3K, and decrease HGFR expression (**Fig 1B**). The combined effect of DISU and fDNA increased HGFR expression, decreased the expression of STAT3, and PI3K, and did not change the expression of CD95 (**Fig 1C**). When DISU and mDNA were co-administered, STAT3 and HGFR expressions were increased, while PI3K and CD95 expressions were decreased (**Fig 1D**).

Because cfDNA treatment affects both TLR9-signaling and the autophagy machinery, we also examined how the effect of concomitant HGFR inhibition and modified DNA treatment is altered by the inhibition of TLR9-signaling or autophagy. By inhibiting TLR9-signaling, co-

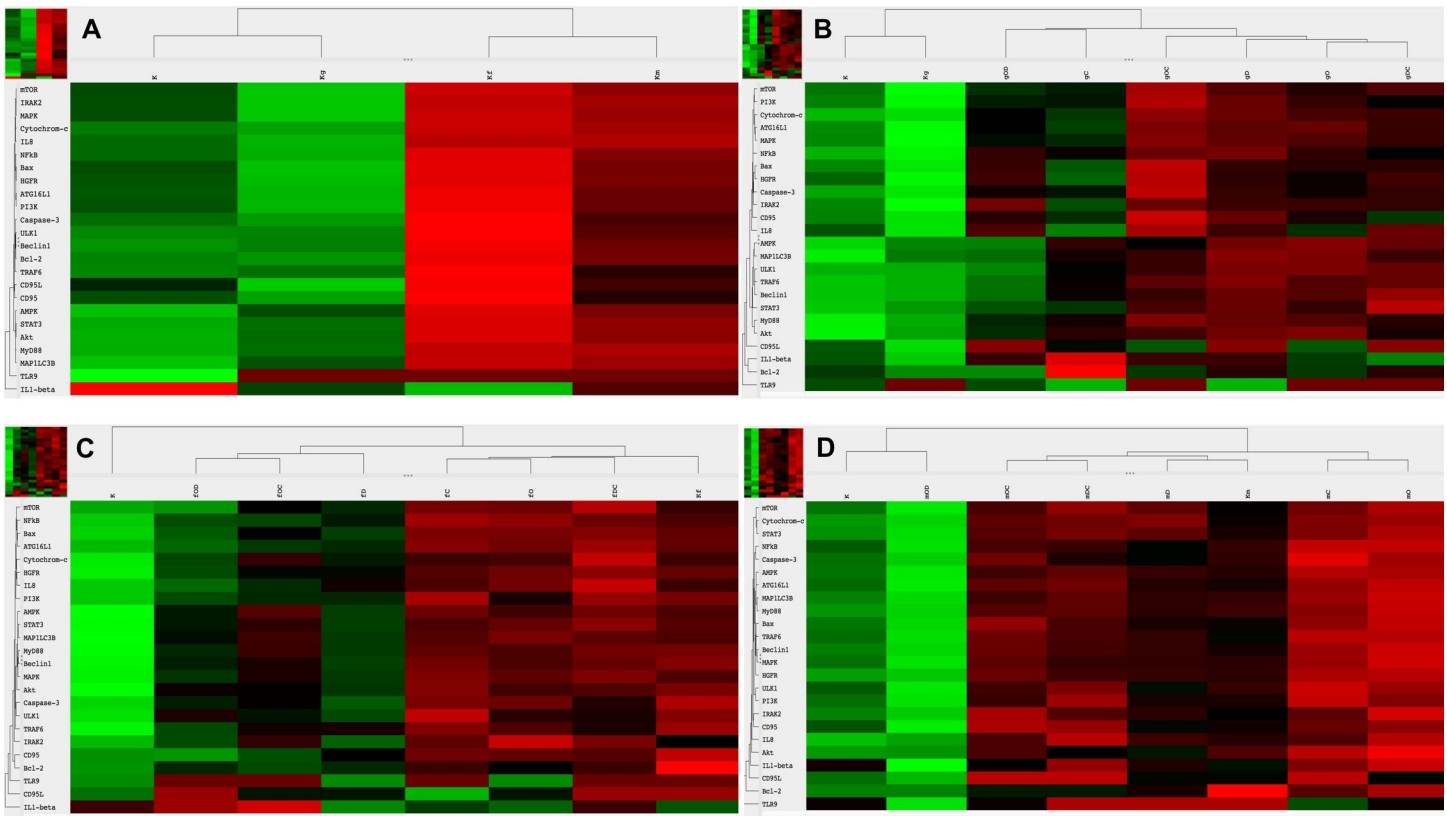

**Fig 1. Heatmap visualization of the NanoString gene expression analyses. A.** Gene expression changes of modified DNA treatments as compared to control, non-treated HT29 cells. Gene expression alterations in HT29 cell after incubation with gDNA (**B.**), fDNA (**C.**), and mDNA (**D.**). g/f/mDNA: genomic / fragmented / hypermethylated deoxyribonucleic acid; O: ODN2088 –TLR9 inhibitor; D: 4,4'Diisothiocyanatostilbene-2,2'-disulfonic acid; C: chloroquine; K: control, non-treated HT29 cells; Kg/f/m: g/f/mDNA treated control HT29 cells; red: overexpression, green: down-regulation.

administration of all types of modified DNAs and DISU decreased the expression of genes involved in canonical and non-canonical C-MET signaling (**Fig 2A**). The inhibition of autophagy together with co-administration of gDNA and DISU did not affect the overexpression of STAT3, on the other hand, the expression of all other genes involved in C-MET-signaling was decreased (**Fig 2A**). Co-administration of DISU and chloroquine with fDNA or mDNA had a pronounced stimulatory effect on the expression of all elements of C-MET-signaling (**Fig 2A**).

As for autophagy-related genes, the effect of combined HGFR inhibition and modified DNA treatments resulted in upregulation of ATG16L1, MAPLC3B, Beclin1 and ULK1 except in relation of fDNA and Beclin1, and mDNA and ULK1, where the expression of these genes was not altered as compared to normal control (**Fig 2B**). Concomitant HGFR inhibition, modified DNA treatments and TLR9 inhibition resulted in downregulation of all autophagy-related genes when treated with gDNA or mDNA, while MAPLC3B, Beclin1, and ULK1 were overexpressed, and ATG16L1 showed no gene expression alteration in case of incubation with fDNA. Combining DISU, chloroquine and modified DNAs led to the upregulation of all autophagy-related genes (**Fig 2C**).

The results of Taqman RT-PCR validated the gene expression alterations detected by NanoString/nCounter analysis. Fold changes (and SD values) of the analyzed gene expressions are summarized in **Fig 3**.

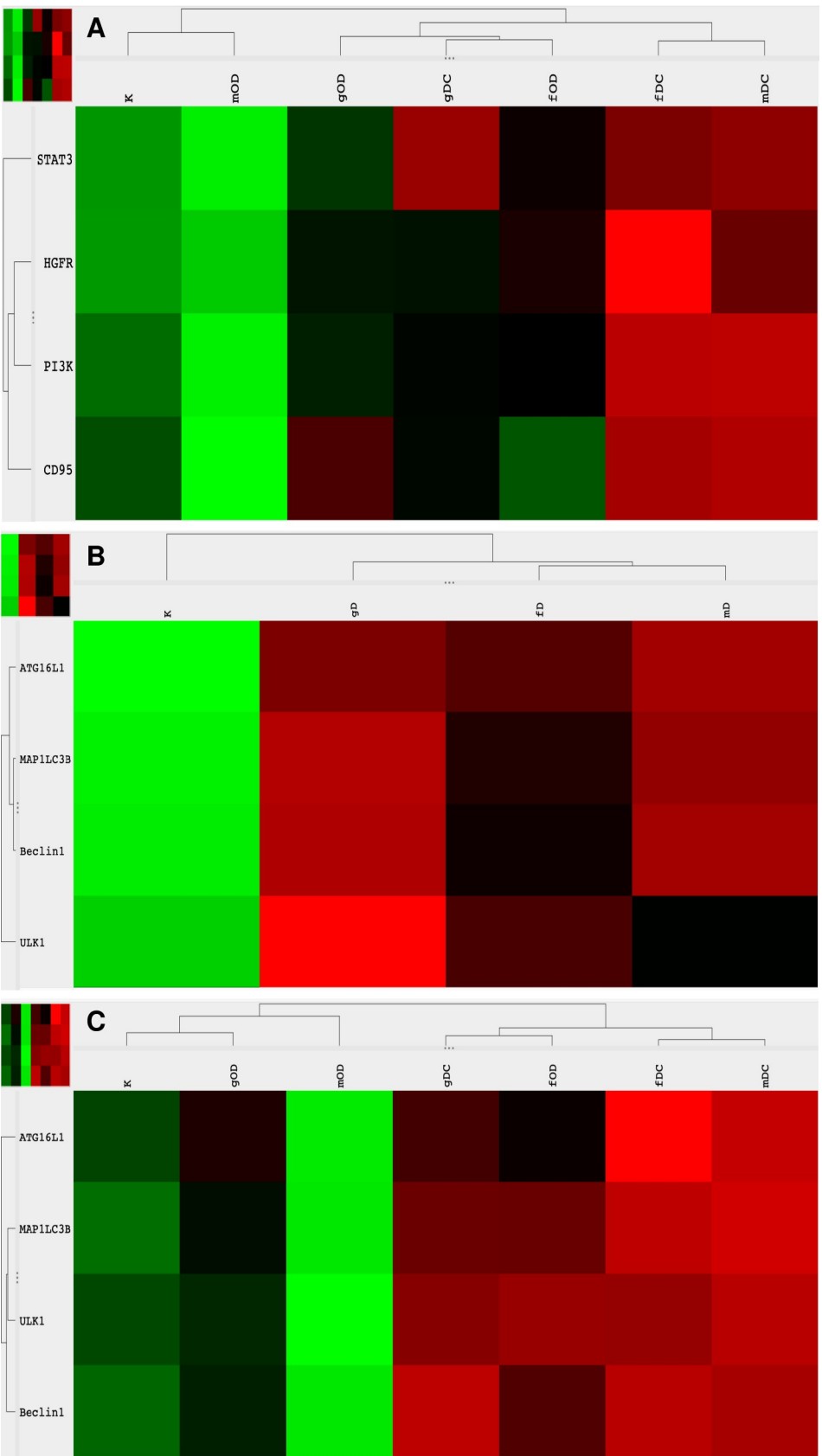

Fig 2. Heatmap visualization of the NanoString gene expression analyses: Alterations in C-MET signaling pathways and autophagy. Gene expression changes of combined treatments with modified DNAs, ODN2088, DISU, or chloroquine. g/f/mDNA: genomic / fragmented / hypermethylated deoxyribonucleic acid; O: ODN2088 –TLR9 inhibitor; D: 4,4'Diisothiocyanatostilbene-2,2'-disulfonic acid; C: chloroquine; K: control, non-treated HT29 cells; red: overexpression, green: down-regulation.

## Immunocytochemistry analyses and WES Simple Western blot results

In selected cases, we performed immunocytochemistry to validate gene expression results at the protein level.

In non-treated control HT29 cells weak to moderate TLR9 immunopositivity was found. Moderate to strong TLR9 protein expressions was detected after incubation with g-, f-, and mDNAs. Regarding HGFR immunocytochemistry weak immunoreaction was found in control and gDNA-treated samples, while strong immunopositivity was observed after f- and mDNA treatments. As for autophagy, f- and mDNAs caused strong upregulation of ATG16L1, Beclin1 and LC3 protein expressions: moderate to strong immunoreactions were detected in these cases as compared to untreated control and gDNA-treated HT29 cells (Fig 4). The immunochemistry results were similar to the NanoString and Taqman gene expression results.

Changes in the amount of LC3B protein in the study groups were consistent with changes in gene expression (NanoString and Taqman) and immunocytochemistry studies. Regarding

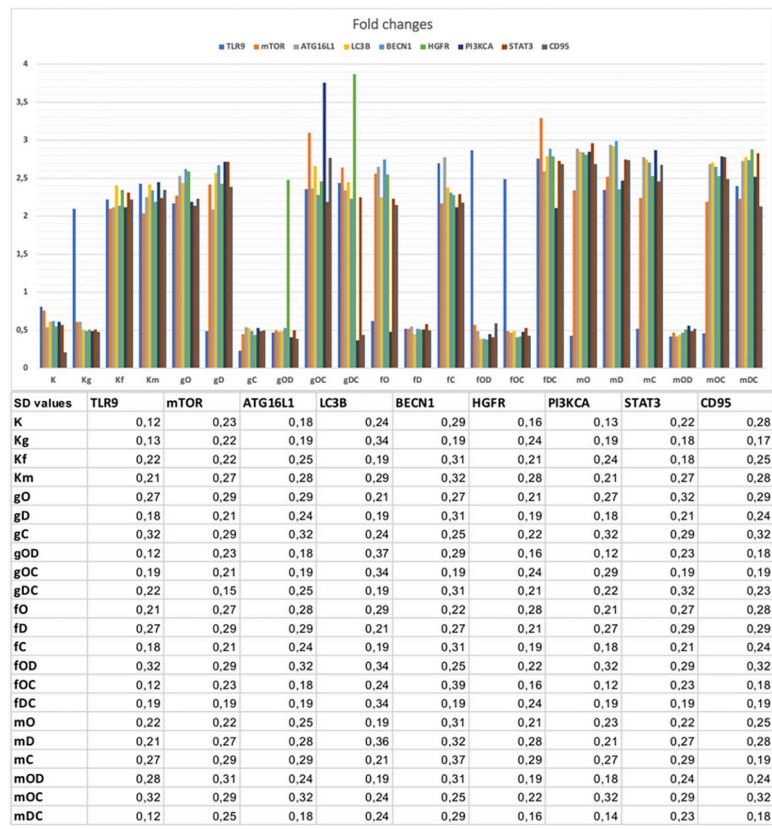

| SD values | TLR9 | mTOR | ATG16L1 | LC3B | BECN1 | HGFR | PI3KCA | STAT3 | CD95 |
|---|---|---|---|---|---|---|---|---|---|
| K | 0,12 | 0,23 | 0,18 | 0,24 | 0,29 | 0,16 | 0,13 | 0,22 | 0,28 |
| Kg | 0,13 | 0,22 | 0,19 | 0,34 | 0,19 | 0,24 | 0,19 | 0,18 | 0,17 |
| Kf | 0,22 | 0,22 | 0,25 | 0,19 | 0,31 | 0,21 | 0,24 | 0,18 | 0,25 |
| Km | 0,21 | 0,27 | 0,28 | 0,29 | 0,32 | 0,28 | 0,21 | 0,27 | 0,28 |
| gO | 0,27 | 0,29 | 0,29 | 0,21 | 0,27 | 0,21 | 0,27 | 0,32 | 0,29 |
| gD | 0,18 | 0,21 | 0,24 | 0,19 | 0,31 | 0,19 | 0,18 | 0,21 | 0,24 |
| gC | 0,32 | 0,29 | 0,32 | 0,24 | 0,25 | 0,22 | 0,32 | 0,29 | 0,32 |
| gOD | 0,12 | 0,23 | 0,18 | 0,37 | 0,29 | 0,16 | 0,12 | 0,23 | 0,18 |
| gOC | 0,19 | 0,21 | 0,19 | 0,34 | 0,19 | 0,24 | 0,29 | 0,19 | 0,19 |
| gDC | 0,22 | 0,15 | 0,25 | 0,19 | 0,31 | 0,21 | 0,22 | 0,32 | 0,23 |
| fO | 0,21 | 0,27 | 0,28 | 0,29 | 0,22 | 0,28 | 0,21 | 0,27 | 0,28 |
| fD | 0,27 | 0,29 | 0,29 | 0,21 | 0,27 | 0,21 | 0,27 | 0,29 | 0,29 |
| fC | 0,18 | 0,21 | 0,24 | 0,19 | 0,31 | 0,19 | 0,18 | 0,21 | 0,24 |
| fOD | 0,32 | 0,29 | 0,32 | 0,34 | 0,25 | 0,22 | 0,32 | 0,29 | 0,32 |
| fOC | 0,12 | 0,23 | 0,18 | 0,24 | 0,39 | 0,16 | 0,12 | 0,23 | 0,18 |
| fDC | 0,19 | 0,19 | 0,19 | 0,34 | 0,19 | 0,24 | 0,19 | 0,19 | 0,19 |
| mO | 0,22 | 0,22 | 0,25 | 0,19 | 0,31 | 0,21 | 0,23 | 0,22 | 0,25 |
| mD | 0,21 | 0,27 | 0,28 | 0,36 | 0,32 | 0,28 | 0,21 | 0,27 | 0,28 |
| mC | 0,27 | 0,29 | 0,29 | 0,21 | 0,37 | 0,29 | 0,27 | 0,29 | 0,19 |
| mOD | 0,28 | 0,31 | 0,24 | 0,19 | 0,31 | 0,19 | 0,18 | 0,24 | 0,24 |
| mOC | 0,32 | 0,29 | 0,32 | 0,24 | 0,25 | 0,22 | 0,32 | 0,29 | 0,32 |
| mDC | 0,12 | 0,25 | 0,18 | 0,24 | 0,29 | 0,16 | 0,14 | 0,23 | 0,18 |

Fig 3. Graphical visualization of the Taqman fold changes. The table displays the related standard deviation results (SD). All SD values were under 0.5 cycle. K: control, non-treated HT29 cells; g/f/mDNA: genomic / fragmented / hypermethylated DNA; O: ODN2088; D: DISU; C: chloroquine.

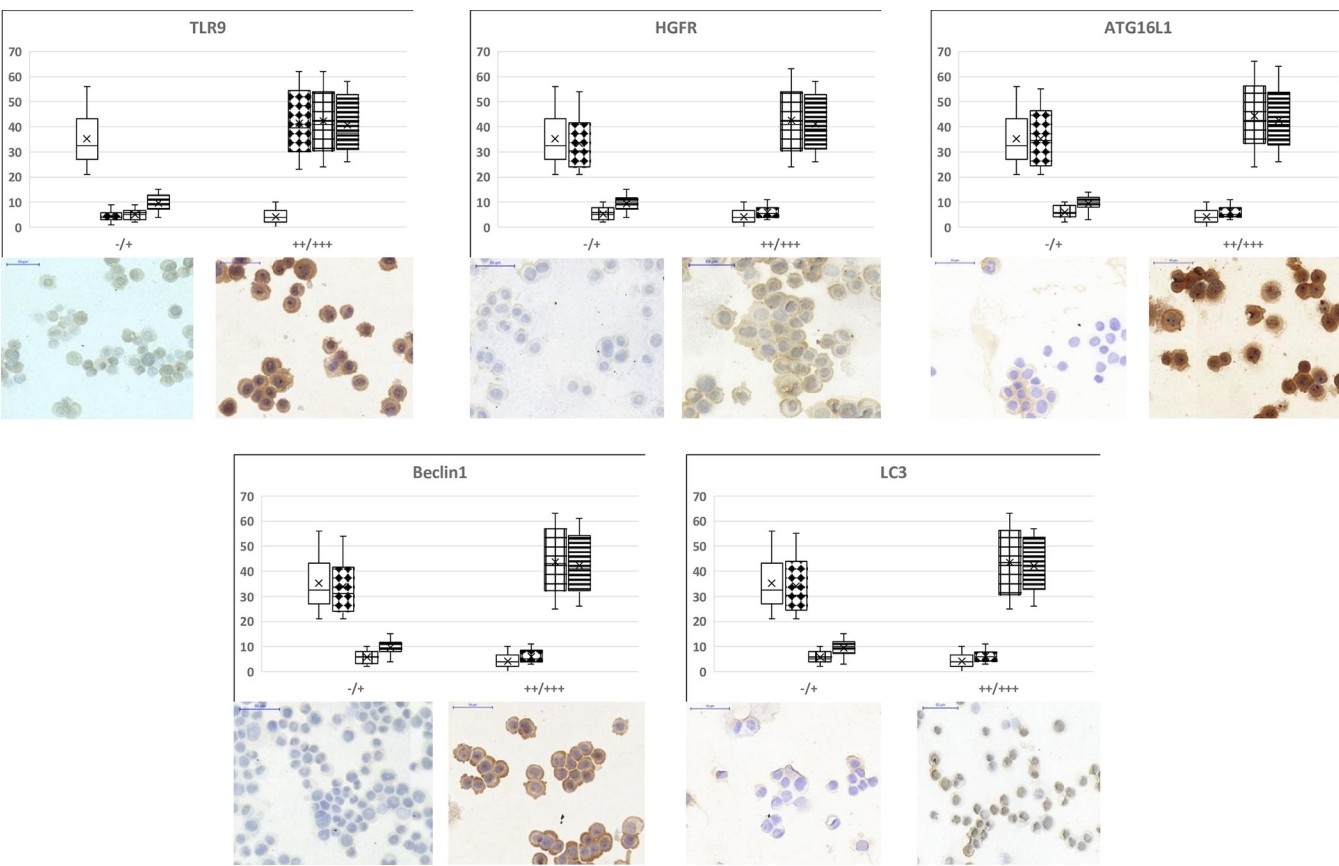

**Fig 4. TLR9, HGFR and autophagy-related protein immunocytochemistry results.** The box and whisker plots represent the one-way ANOVA results of immunocytochemistry analyses. The percentage of non-immunoreactive and weakly immunopositive ("-/+"), as well as moderately and strongly immunopositive ("++/+++") HT29 cells within the treatment groups was visualized. Under the plots representative "-/+" and "++/+++" image inserts can be seen. Scale bars represents 50 μm. Empty boxes: control, non-treated cells; diamond dots boxes: gDNA treatment; square grid boxes: fDNA treatment; striped boxes: mDNA treatment.

autophagy, the levels of LC3B and p62 proteins also show that the inhibitory effect of chloroquine is enhanced by the combination of DNA treatments (g, f, m) and DISU, i.e., they result in enhanced inhibition of autophagy. Inhibition of autophagy, in turn, leads to an accumulation of protein levels by reducing degradation of LC3B and p62 proteins. The results of the WES Simple Western Blot can be seen in **Fig 5**.

## Transmission electron microscopy

Control, non-treated, metabolically active HT29 cells (3 ± 1 pieces/cell), similarly to chloroquine treated controls (4 ± 1.5 pieces/cell), displayed autophagic vacuoles (AVs) in the cytoplasm indicating macroautophagy. The frequency of AVs in DISU (6 ± 1.8 pieces/ cell) and ODN2088 (7 ± 1.4 pieces/cell) control cells was higher as compared to control. Incubation with gDNA resulted in the appearance of a more intense macroautophagy (6 ± 2 pieces/cell), and co-administration of ODN2088 (9 ± 1.2 pieces/cell), DISU (7 ± 2 pieces/cell), or chloroquine (7 ± 1.6 pieces/cell) also favored the presence of an intense autophagy. Following fDNA administration (5 ± 1.8 pieces/cell), not only AVs but also multivesicular bodies (MVBs) appeared. The combined effect of ODN2088 and fDNA (12 ± 2 pieces/cell) intensified the process of autophagy (fDNA+DISU: 6 ± 1.4 pieces/cell; fDNA+ chloroquine: 4 ± 2.3 pieces/cell).

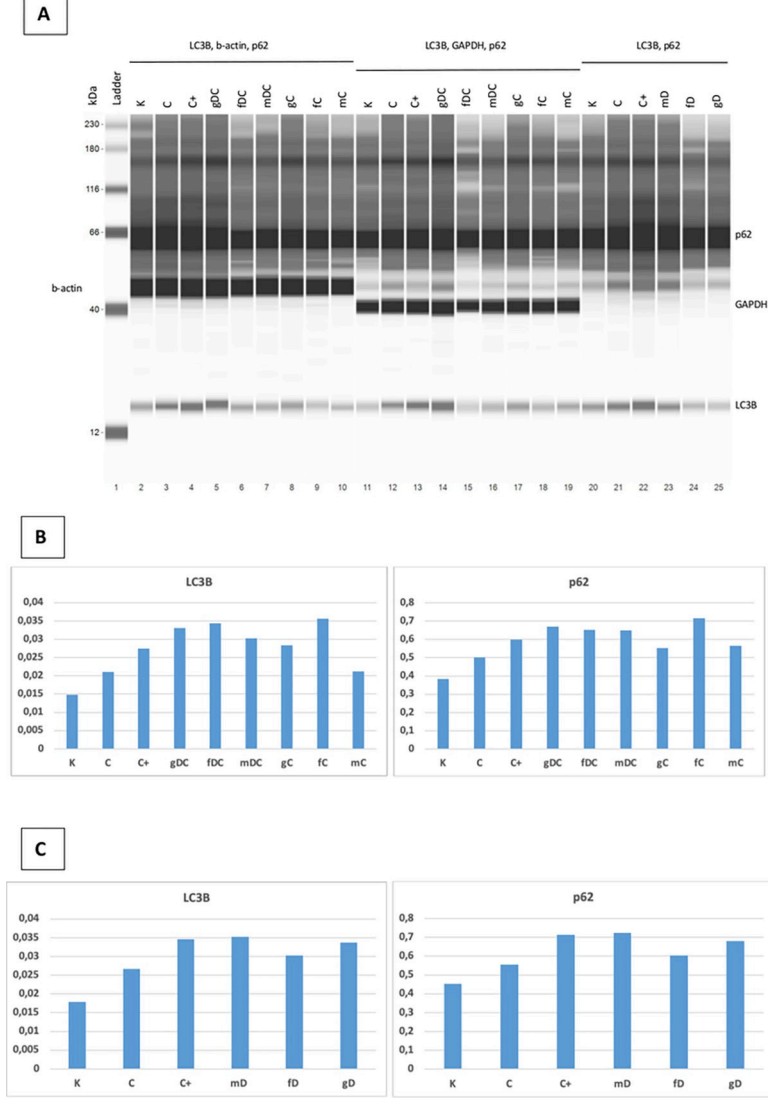

**Fig 5. Results of the p62/sequestrome 1 and LC3B WES Simple Western blot. A.** Representative blot images. **B** and **C.** Graphical representation of the protein expressions: the area of the tested proteins was multiplied by the values of the β-actin area. K: control, non-treated HT29 cells; C: chloroquine (10 μM); C+: chloroquine (50 μM); g: genomic DNA; f: fragmented DNA; m: hypermethylated DNA; D: DISU; C: chloroquine (10 μM).

However, co-administration of fDNA and any inhibitor made the cells disintegrated. Following incubation with mDNA (7 ± 1.3 pieces/cell) the cell structure also became disorganized along with chromatin condensation and blebbing. mDNA in combination with chloroquine (5 ± 1.6 pieces/cell) resulted the appearance of MVBs. mDNA together with DISU (7 ± 1.4 pieces/cell), however, enhanced cell survival, and the activated macroautophagy apparently contributed to maintain cellular fitness. mDNA together with ODN2088 resulted in the lowest number of AVs (2 ± 1.3 pieces/cell).

Thus, the presence of autophagy was observed in each group of HT29 cells, but to a varying degree. The representative microstructural changes can be seen in **Fig 6**.

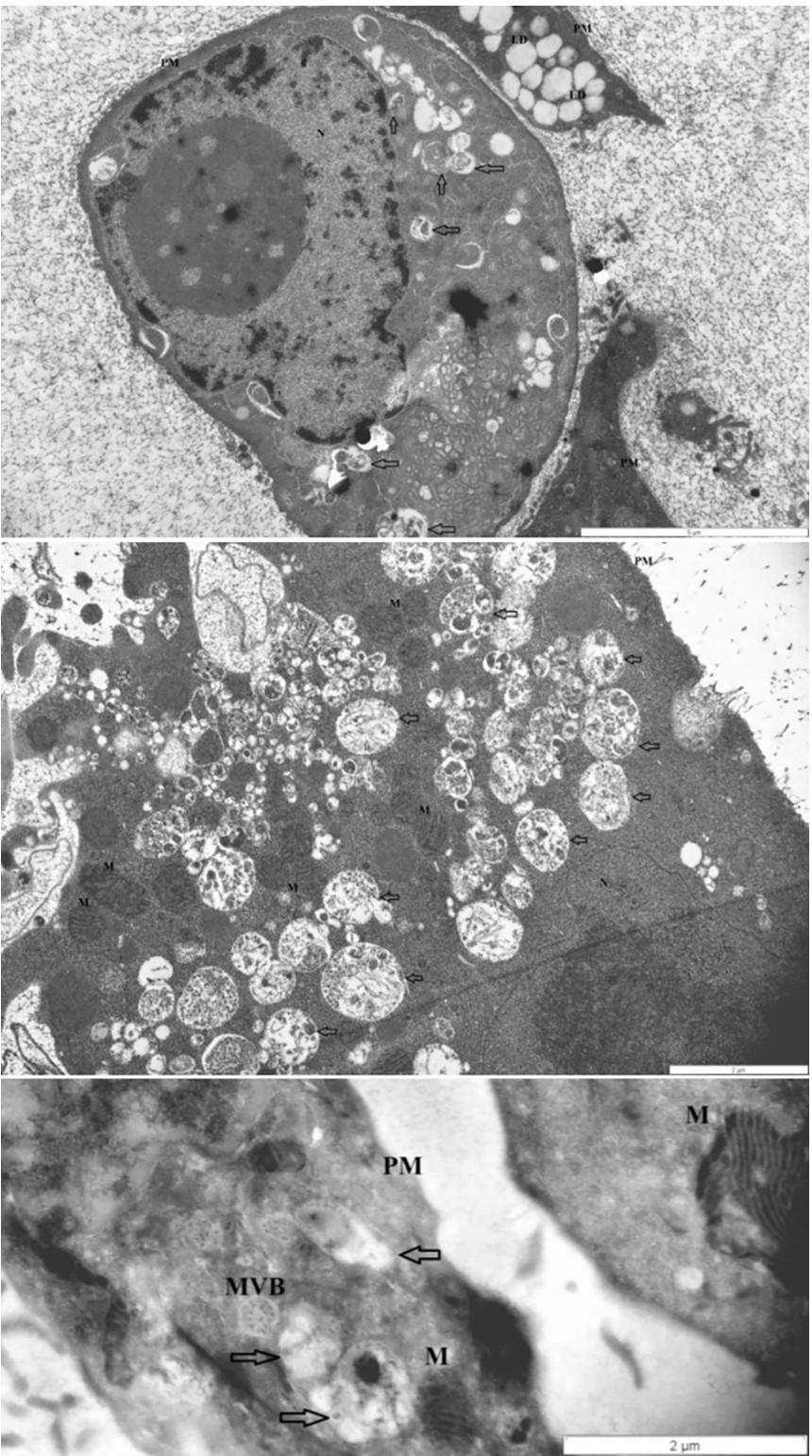

**Fig 6. Transmission electron microscopy results.** The representative images highlight the autophagy-related structural changes in HT29 cells (from top to down: disorganized nucleus with chromatin condensation plus autophagic vacuoles; single and aggregated autophagic vacuoles; multivesicular body). Arrows: autophagic vacuoles; MVB: multivesicular body; PM: plasma membrane; N: nucleus; M: mitochondrium; L: lipid droplet.

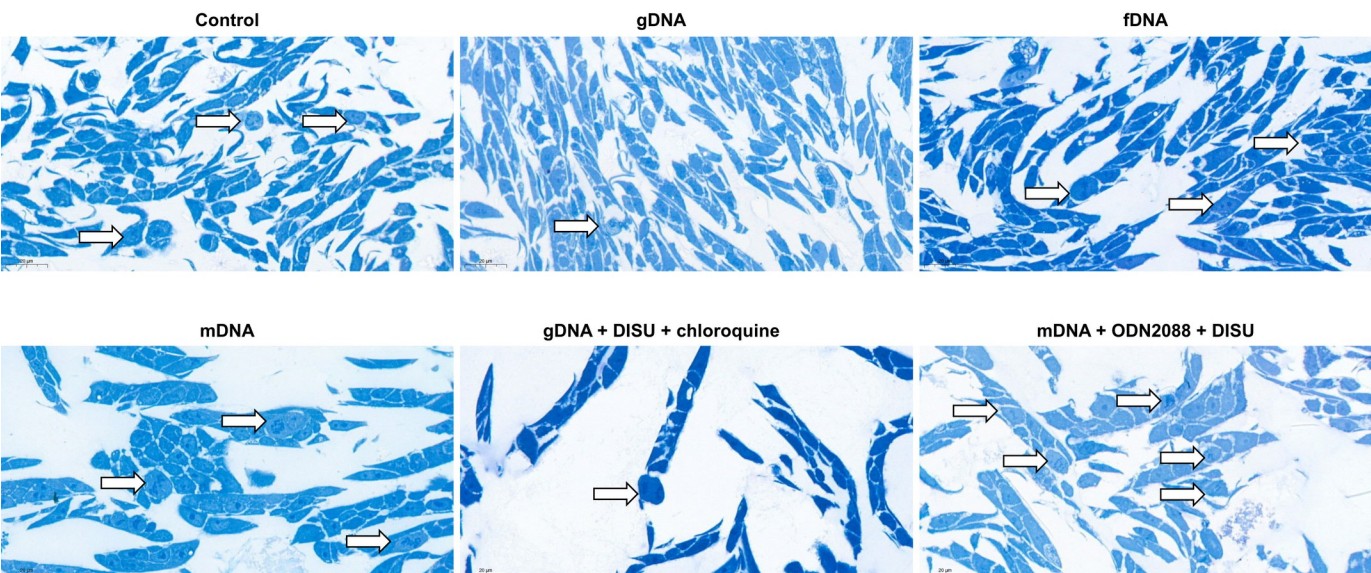

**Fig 7. Signs of proliferative activity in HT29 cells.** In case of gDNA, DISU and chloroquine co-administration, the number of cell divisions was decreased. After mDNA, ODN2088 and DISU administration, the proliferation activity of HT29 cancer cells increased. Arrows indicate cell divisions; scale bar represents 20 μm.

### Semithin sections

To investigate whether the decrease in cell numbers after treatments with modified self-DNAs and/or TLR9, HGFR, or autophagy inhibitors was due to low proliferation activity or increased cell death, semi-thin sections were also examined in selected cases. In case of incubation with g-, f-, or mDNAs the incidence of proliferation was proportional to the cell numbers obtained. When gDNA was co-administered with DISU and chloroquine, remarkably reduced proliferative activity was observed. In contrast, after combining mDNA with ODN2088 and DISU, higher proliferative activity was detected (**Fig 7.**).

## Discussion

In this study, we sought to answer how the combination of intact or modified tumorous self-DNA treatment with the inhibition of TLR9 signaling, autophagy, and/or HGFR signaling affects the viability and proliferation of HT29 cells.

First, we determined the effect of self-DNA-induced TLR9 signaling modulation on HT29 cell survival. The existence of cell-free nucleic acids (including cfDNA sequences) in human blood, urine, saliva or feces is a known fact [31]. The methylation status or fragmentation of cfDNAs may carry information about their source [32, 33]. In terms of their origin, cfDNAs fall into several categories. Endogenous cfDNA sequences are derived from tissues and cells, while exogenous ones are derived primarily from the host microbiome, infectious agents, fetus, and food [33–36].

Toll-like receptors (TLRs) are innate immune receptors [37]. TLR9 is capable of detecting DNA from both endogenous and exogenous sources [37]. We have previously demonstrated that the structural modifications of self-DNA (i.e., methylation status and fragment length) plays a significant role in activation of the TLR9-mediated signaling pathways [23, 38].

In HT29 cells the constitutive expression of TLR9 mRNA was described [39]. Their basal TLR9 mRNA expression is low, whereas TLR9 expression is increased by incubation with CpG-ODN or tumorous self-DNA [38, 39]. We also found that TLR9 gene expression was

increased in groups of cells treated with genomic, fragmented, or hypermethylated self-DNAs as compared to the non-treated, control HT29 cells.

Autophagy can be triggered by CpG-oligodeoxynucleotides in tumor cell lines (e.g.: colon, breast, and prostate cancers) via a TLR9-dependent manner [27]. Signals linking TLRs and autophagy could be both the altered glyceraldehyde-3-phosphate dehydrogenase (GAPDH) gene expression and the generation of reactive oxygen species (ROS) [40–42]. There are several shared features between TLR9 and autophagy pathways, such as their effects on cell survival and death, their role in innate immunity, the induction of MHC class II antigen presentation, their interactions in endosomes, the positive effect of class III PI3K on their signaling, or their common inhibitors (e.g., hydroxychloroquine, 3-methyladenine, bafilomycin A1) [27]. We have recently provided evidence for a close interplay between TLR9-signaling and autophagy response with remarkable influences on survival in HT29 cells subjected to modified self-DNA treatments [23].

In our present study, modified self-DNA treatments altered the metabolic activity and proliferation of HT29 cells to varying degrees. Interestingly, in the case of gDNA, this was due to a decrease in the expression level of all examined genes, while in the case of fragmented and hypermethylated DNAs, an increase was observed. This could be due to the fact that the examined elements of TLR9-signaling pathway may exhibit both pro- and anti-survival effects [23, 43–47], and differently modified self-DNA sequences may activate this complex signaling pathway [23].

In cells treated with modified self-DNAs, blocking TLR9 signaling increased the metabolic activity in all cases. In terms of cell division, incubation with genomic and hypermethylated DNAs decreased, while fragmented DNA treatment slightly increased cell proliferation. TLR9 signaling inhibition, however, counteracted the effect of self-DNA treatments. In the background of this observation, beside the role of distinct levels of TLR9 signaling activation the different expression of pro- and anti-apoptotic genes can be assumed [48–52].

In the following steps, the effect of changes in the interaction of HGFR and TLR9 signaling pathways on HT29 cell survival was investigated. Few information is available on this complex signaling crosstalk. Recently, it has been demonstrated that the activation of TLR2 and TLR5 in epithelial cells induces phosphorylation of RTKs involved in epithelial repair, growth and carcinogenesis. Besides all members of epidermal growth factor receptor (EGFR) family, other RTKs, including HGFR can be activated by TLR stimulation [27]. TLR-MyD88 signaling and chemotactic stimuli could activate extracellular signal-regulated kinases (ERKs). ERKs can be activated by Ras, which can be activated by growth hormones via RTKs. Specifically, in T. gondii-infected macrophages, both TLR-MyD88-dependent and TLR-MyD88-independent ERK activation has been described [53]. Since MyD88 is a key element of TLR9 signaling pathway, one cannot be ruled out that there is a molecular relationship between the TLR9 and HGFR signaling pathways.

Here we found that MyD88 expression tends to increase with self-DNA treatment, and this effect is not affected or further enhanced by HGFR inhibition. The overexpression of MyD88 was associated with enhanced HT29 cell proliferation. It has been recently found, that MyD88 displays anti-apoptotic functions in colon carcinoma cells through the Ras/Erk, but not the NF-κB pathway [54]. We observed that the expression of caspase-3 also changed similarly to that of MyD88. Consequently, the effect of inhibited apoptosis-induced compensatory cell proliferation, in which caspases (e.g., caspase-3) take an important role may partly explain the observed alterations of HT29 cell proliferation [55].

We also examined how the interaction of TLR9 and HGFR signaling affects autophagy and HT29 cell proliferation. Strict control of RTK trafficking is crucial for normal homeostasis. In human cancers, RTKs avoid entry into degradation pathways [56]. Signaling cascades

downstream to growth factor RTKs, changes in energy levels as well as nutrient availability have been shown to control autophagy [57]. LC3C-mediated autophagy was found to selectively regulate the HGF/HGFR-stimulated migration and invasion in HeLa cancer cells [28]. Regarding the interplay between autophagy machinery and HGFR signaling in colorectal cancer cell lines, it has been recently found that mammalian target of rapamycin complex (mTORC)1-independent basal autophagy positively modulates phosphorylation levels of several RTKs, including HGFR. Additionally, it has been shown that genetic suppression of basal autophagy decreases mTORC2-mediated activation of Akt, but does not affect mTORC1 activity. It has also been demonstrated that autophagy positively mediates the phosphorylation of HGFR via regulation of mTORC2 since reduced mTORC2 activation in autophagy-defected cells was responsible for the impaired HGFR phosphorylation [29].

Although all types of modified self-DNA treatment increased the expression of autophagy-related genes, cell proliferation was only enhanced after fDNA administration. The combination of DISU and DNA led to the accumulation of p62 and LC3B proteins, which means that autophagy was impaired. With respect to cell proliferation, the combined effect of DISU and DNA was the opposite to that of DNA alone. Recently, a complex bidirectional relationship between autophagy-master regulator kinases and autophagy-related proteins has been unraveled [58]. HGFR and β1-integrin colocalize with Beclin1 and/or LC3B-positive compartments and a pool of phosphorylated extracellular signal-regulated protein kinase (ERK)1/2 localize along with HGFR in autophagy-related endomembranes following HGF stimulation [59]. Furthermore, HGFR has been found to partially colocalize with LC3B-positive perinuclear vesicles, which may affect its phosphorylation, since chloroquine-mediated accumulation of autophagosomes increases HGFR phosphorylation only in autophagy-proficient circumstances [29]. For that reason, autophagic vesicles could represent signaling platforms whereby HGFR phosphorylation is controlled via mTORC2 [58].

The most pronounced reduction in cell proliferation was achieved with the concomitant use of gDNA, DISU, and chloroquine. In this case, the overexpression of STAT3, which is involved in C-MET non-canonical signaling, was observed. STAT3 activity in colon carcinoma cells is triggered via interleukin-6 (IL6) or through a constitutively active STAT3 mutant promoted cancer cell multiplication [60]. Based on these, STAT3 has a stimulating effect on cell proliferation. Activation of TLR9 also induces IL6 production [61], which is favorable for STAT3 activation. On the other hand, LC3B was also overexpressed in this group of HT29 cells. In case of LC3B upregulation, the threshold of LC3B activation increases, which can dictate pro-apoptotic function [62]. In our case, it cannot be ruled out that the proliferative stimulating effect of STAT3 expression was outweighed by the inhibitory effect of LC3B, which finally resulted in remarkably decrease of cell proliferation. Furthermore, we observed the accumulation of LC3B and p62 proteins in the gDC group. This suggests that this combination of treatments resulted in defective autophagy. Excessive accumulation of p62 in tumor cells is characterized by cell cycle initiation, inhibition of apoptosis, and thus enhanced proliferation ability [63]. However, in a recent study [64], high LC3B dot-like/high p62 dot-like-cytoplasmic protein expression (indicative of impaired autophagy) was associated with the best prognosis in CRC patients. These data suggest an association between inhibition of autophagy and decreased cell proliferation. Although, it should not be forgotten that chloroquine has been reported to kill cancer cells independently of autophagy inhibition [65–67], the 10 μM chloroquine we used on its own effectively inhibited autophagy without affecting the proliferation of HT29 cells (S2 Fig). Based on these, the combined effect of gDNA and DISU used in addition to chloroquine can be assumed in the background of the decrease in cell proliferation in the gDC group.

The induction of cytoprotective autophagy in HGFR expressing cells upon HGFR inhibition or combined HGFR and autophagy inhibition was found to result in significantly decreased cell viability in gastric adenocarcinoma cells [68]. It has been demonstrated that autophagy induction along with mTOR and ULK1 dephosphorylation upon HGFR inhibitor treatment could be alleviated by HGF and mTOR agonist MHY1485, implying that autophagy was initiated by HGFR inhibitors via Met-mTOR-ULK1 molecular cascade. Interestingly, in the presence of inhibited autophagy HGFR inhibitors further suppressed cell survival and tumor growth in Met-amplified cancer cells. Hence, these results suggest that HGFR-mTOR-ULK1 cascade is responsible for HGFR inhibitor-mediated autophagy, and HGFR inhibitors combined with autophagy inhibitors could be a promising choice to treat Met-amplified cancers [15].

The highest extent of cell proliferation was observed when mDNA, ODN2088 and DISU were co-administered. In this case, decreased expression of both canonical and non-canonical HGFR signaling pathways was observed, interestingly, together with the down-regulation of autophagy-related genes. Reduced autophagy has been demonstrated to increase cell proliferation via an as-yet-unidentified mechanism [69–71]. It was demonstrated that genetic silencing of key autophagy proteins (e.g., Beclin 1, Ambra 1) in mice can lead to increased cell proliferation [71]. The possibility that the combination of treatments used had an epigenetic regulatory effect on the autophagy/cell proliferation interaction cannot be ruled out. Furthermore, regarding the assayed genes, the overexpression of CD95L and IL8 was primarily observed. CD95L can induce apoptosis by binding CD95, its cognate receptor. More recently, it has been discovered that CD95L can also induce proliferation, differentiation and cell migration [72]. IL8 was found to stimulate cell proliferation in non-small cell lung cancer through EGFR transactivation [73]. An intensive cross-talk between HGFR and EGFR is existing [11]. Based on these results, intense proliferation in mOD HT29 cell group may be due not only to overexpression of CD95L and IL8, but HGFR/EGFR cross-signaling may also play a role in. In addition to inhibiting autophagy, TLR9 and/or HGFR signaling pathways, the use of modified tumor self-DNAs allows the development of novel anticancer therapies. The effects of the co-administration of these agents should also be investigated in other tumor cell lines and animal models.

Recently, it has been demonstrated that multivesicular body (MVB)-like small extracellular vesicle complexes can be released by HT29 cells in the absence of stromal cells [74]. In this study, the observed ultrastructural alterations call attention to the role of autophagy in cell protection or even in promoting cell death. In the cell groups where the presence of MVBs was detected, the expression of Beclin1 and PI3K genes was increased. This suggests that autolysosomal degradation is also likely to be present following the formation of amphisomes through the interconnection of autophagosome and multivesicular body pathways [75]. Amphisome serves as a prelysosomal compartment in which both the endocytic and autophagic pathways converge [76, 77]. The contents of amphisomes could have multiple fates, such as extracellular release or lysosomal degradation. Both exosome biogenesis and autophagy display pivotal roles in maintaining cellular homeostasis and enhancing stress tolerance [78]. Influencing these functions for cancer cells may allow the identification of realistic therapeutic targets.

## Conclusion

In summary, in this study we aimed to assess the complex interaction of HGFR, TLR9 signaling and autophagy inhibition on the survival and proliferation of HT29 colon cancer cells upon modified tumorous self-DNA treatments. We found that the decrease of cell proliferation depends on the type of DNA modification. The use of TLR9 blocking has reversed this

effect. MyD88 expression was found to slightly increase with self-DNA treatments. The over-expression of MyD88 was associated with enhanced HT29 cell proliferation, and the expression of caspase-3 also changed similarly to that of MyD88. Consequently, incubation with modified self-DNAs could suppress the apoptosis-induced compensatory HT29 cell proliferation. All types of modified self-DNA treatments increased the expression of autophagy-related genes. DISU inhibited the proliferation-reducing effect of genomic and hypermethylated DNAs, and displayed the opposite effect when fragmented DNA was used. The most pronounced reduction in cell proliferation was achieved with the concomitant use of gDNA, DISU, and chloroquine. In this case, the proliferation stimulating effect of STAT3 overexpression could be outweighed by the inhibitory effect of LC3B, indicating the putative involvement of HGFR-mTOR-ULK1 molecular cascade in HGFR inhibitor-mediated autophagy. The highest extent of cell proliferation was observed when the co-administration of mDNA, ODN2088 and DISU was performed. In this case, decreased expression of both canonical and non-canonical HGFR signaling pathways and autophagy-related genes was present. The ultrastructural changes we observed also support the context-dependent role of HGFR inhibition and autophagy on cell survival and proliferation. Further investigation of the influence of the studied signaling pathways and cellular processes can provide a basis for novel, individualized anti-cancer therapies.

## Supporting information

**S1 Fig.** Changes in the metabolic activity (blue) and proliferation (orange) of the studied cell groups under the influence of each treatment combination. The red star indicates the lowest (group gDC), while the red triangle indicates the highest proliferative activity (mOD group). g/f/mDNA: genomic/fragmented/hypermethylated deoxyribonucleic acid; ODN: CpG oligonucleotide; DISU: 4,4'Diisothiocyanatostilbene-2,2'-disulfonic acid; C: chloroquine; SD: standard deviation.
(TIF)

**S2 Fig. Proliferation inhibitory effect of different concentrations of chloroquine in HT29 and DLD1 colon adenocarcinoma cells.** Based on our preliminary experiments, chloroquine treatment at a concentration of 10 μM effectively inhibited autophagy without significantly affecting the proliferation of HT29 cells after 72h of incubation.
(TIF)

**S1 Table. On-way ANOVA and Tukey HSD test results.** Regarding HT29 cell viability and cell number data, the results of the statistical analyses with one-way ANOVA and Tukey HSD test are displayed.
(PDF)

## Acknowledgments

The authors would like to acknowledge Eszter Krisztina Göttl and Nikolett Dóczi (Department of Anatomy, Histology and Embryology, Semmelweis University), Gabriella Kónyáné Farkas (Heim Pál National Institute of Pediatrics), and Anika Scott for skillful technical assistance.

## Author Contributions

**Conceptualization:** Bettina Bohusné Barta, Anna Sebestyén, Ferenc Sipos, Györgyi Műzes.

**Data curation:** Bettina Bohusné Barta, Anna Sebestyén, Ferenc Sipos, Györgyi Műzes.

**Formal analysis:** Bettina Bohusné Barta, Anna Sebestyén, Ferenc Sipos, Györgyi Műzes.

**Funding acquisition:** Ferenc Sipos, Györgyi Műzes.

**Investigation:** Bettina Bohusné Barta, Ágnes Simon, Lőrinc Nagy, Titanilla Dankó, Regina Eszter Raffay, Viktória Zsiros, Anna Sebestyén, Ferenc Sipos, Györgyi Műzes.

**Methodology:** Bettina Bohusné Barta, Ágnes Simon, Lőrinc Nagy, Titanilla Dankó, Regina Eszter Raffay, Gábor Petővári, Viktória Zsiros, Anna Sebestyén, Ferenc Sipos.

**Project administration:** Bettina Bohusné Barta, Ágnes Simon, Lőrinc Nagy, Titanilla Dankó, Regina Eszter Raffay, Ferenc Sipos.

**Resources:** Ferenc Sipos, Györgyi Műzes.

**Software:** Lőrinc Nagy, Titanilla Dankó, Ferenc Sipos, Györgyi Műzes.

**Supervision:** Bettina Bohusné Barta, Regina Eszter Raffay, Anna Sebestyén, Ferenc Sipos, Györgyi Műzes.

**Validation:** Bettina Bohusné Barta, Ágnes Simon, Titanilla Dankó, Regina Eszter Raffay, Gábor Petővári, Anna Sebestyén, Ferenc Sipos, Györgyi Műzes.

**Visualization:** Bettina Bohusné Barta, Lőrinc Nagy, Titanilla Dankó, Gábor Petővári, Viktória Zsiros, Ferenc Sipos, Györgyi Műzes.

**Writing – original draft:** Bettina Bohusné Barta, Ferenc Sipos, Györgyi Műzes.

**Writing – review & editing:** Ferenc Sipos, Györgyi Műzes.

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
