## [Decision Letter · Decision Letter 0]

20 Jan 2022

PONE-D-21-29031Survival of HT29 cancer cells is influenced by hepatocyte growth factor receptor inhibition through modulation of self-DNA-triggered TLR9-dependent autophagy responsePLOS ONE

Dear Dr. Sipos,

Thank you for submitting your manuscript to PLOS ONE. After careful consideration, we feel that it has merit but does not fully meet PLOS ONE’s publication criteria as it currently stands. Therefore, we invite you to submit a revised version of the manuscript that addresses the points raised during the review process.

Academic Editor's comments: While the aim of the study is relevant, there are several important issues requiring authors' attention.

1) The main drawback of the study is the lack of autophagic flux assessment. While the Methods section contains the "Assessing autophagic flux" part, it describes how the cells were treated with chloroquine, but not how the flux was actually assessed. Since the autophagic flux was also not determined in their previous publication (Pathol Oncol Res 2019;25:1505), the authors are advised to perform the flux measurement experiments (e.g. immunoblot assessment of LC3 conversion in the presence of lysosomal inhibitors and/or levels of autophagy-selective targets such as SQSTM1/p62) in accordance with the current guidelines (Autophagy 2021;17:1-382. doi: 10.1080/15548627.2020.1797280).

2) The doubling time of HT29 cells is approx. 24 h. Therefore, it is unclear how the proliferation could be suppressed by almost 90% (Table 2, gDC treatment) after only 4 h of incubation? Since the authors observed that some treatments caused cells to "disintegrate", the observed suppression is more likely due to cell death. This issue should be clarified. 

3) Either Table 2 or Fig. 1 should be omitted, as they present the same data.

4) Table 2/Fig. 1: ANOVA should be used to compare different treatments where appropriate, and the significant differences could be indicated by compact letter display. Also, the authors should state which post-hoc test (if any) was used for multiple comparisons.

5) Fig. 2 and Fig. 3: Gene expression data should be presented as mean ± SD  and the observed differences statistically analyzed. 

6) The concentration of chloroquine (1 μM) seems unusually low for autophagy inhibition. The authors' own data show 3 ± 1 vs. 4 ± 1.5 autophagic vesicles in control vs. chloroquine-treated cells, indicating no significant inhibition of basal autophagy, and the similar results (no additional accumulation of autophagic vesicles and thus no autophagy inhibition) were observed when choloroquine was combined with other treatments. This makes difficult any interpretation of the role autophagy could possibly have in the observed effects. While I am aware of the extent of required additional experimentation, the experiments should be repeated with the chloroquine concentration that inhibits autophagy. An alternative possibility would be to describe chloroquine effects as autophagy-independent, in which case the authors must provide a rationale for using the drug at the concentration unable to block autophagy/

7) Chloroquine has been reported to kill cancer cells independently of autophagy inhibition (Autophagy 2017;13:955, Graefes Arch Clin Exp Ophthalmol 2015;253:2309, Eur J Pharmacol 2019;859:172540, Pharm Res 2012;29:2249), which should be acknowledged and discussed.

8) The effects of multiple treatments on cell proliferation should be explained more clearly. Namely, the authors seem to suggest that the anti-proliferative effect of self-DNA was partly mediated by autophagy induction, and that cell proliferation in various treatment is inversely correlated with autophagy. Therefore, the increase in autophagy gene expression and further block in cell proliferation by chloroquine is somewhat counterintuitive and requires a thorough explanation.

9) Lines 308-310: What is the difference between "moderate upregulation" and "mild overexpression"? If there is none, please rewrite the sentence.   

10) The introductory sentence of the Discussion section should be revised to increase clarity. "In this study, we attempted to answer how HGFR inhibition modulates the effect of tumor-derived self-DNA on TLR9 signaling and autophagy response by examining the metabolic activity and proliferation of HT29 colon cancer cells." One cannot examine the effects on autophagy by measuring metabolic activity and proliferation.

Reviewer 1:

This is an interesting study and authors demonstrated how HGFR inhibition modulates the effect of tumor derived self-DNA on TLR9 signaling and autophagy response by examining the metabolic activity and proliferation of HT29 colon cancer cells that can be considered as continuation of authors previous study entitled, “Modified Genomic Self-DNA Influences In 608 Vitro Survival of HT29 Tumor Cells via TLR9- and Autophagy Signaling” (Ref. 21). The authors have collected a unique dataset using cutting edge methodology. The paper is generally well structured.

- However, in my opinion author can prove the mechanism in another colon cancer cell line with an inducible or overexpressed system beside HT29 cells with constitutive expression of TLR9.

- The authors should execute more molecular biology based experiments such as rtPCR or western blot of most important genes or proteins respectively to show the signaling pathway and metabolic activity based mechanism.

- In the methodology part, some lines are exactly imitated from Ref.21. The authors should take care to avoid plagiarism, specially “HT-29 cell treatment”, “C-MET, TLR9 and autophagy immunocytochemistry”, and “Transmission electron microscopy for evaluation of autophagy” portion. 

- The authors may also include the following reference on Autophagy in the introduction section along with 13, i) A novel betulinic acid analogue ascertains an antagonistic mechanism between autophagy and proteosomal degradation pathway in human colon carcinoma cells (HT29). BMC Cancer, 2016, 16:23, 1-19.(DOI: https://doi.org/10.1186/s12885-016-2055-1)

- The authors may also include the following reference on Aptamer targeted therapy for colon cancer, in the introduction section along with 19, ii) Aptamer-Conjugated Apigenin Nanoparticles To Target Colorectal Carcinoma: A Promising Safe Alternative of Colorectal Cancer Chemotherapy, ACS Appl. Bio Mater, 2018. (DOI: 10.1021/acsabm.8b00441)

We look forward to receiving your revised manuscript.

Kind regards,

Vladimir Trajkovic

Academic Editor

PLOS ONE

Journal Requirements:

2. PLOS ONE does not permit references to unpublished data; therefore, we request that you either include the referenced data or remove the instances of "data not shown," "unpublished results," or similar.

3. Thank you for stating the following in the Acknowledgments/ Funding Section of your manuscript: 

StartUp Program of Semmelweis University Faculty of Medicine (CO No.: 11720, Ikt.sz.: 5127/AOKGIE/2018; SE10332470)

The study was funded by the StartUp Program of Semmelweis University Faculty of Medicine (CO No.: 11720, Ikt.sz.: 5127/AOKGIE/2018; SE10332470) awarded to FS and GM.

Reviewers' comments:

Reviewer's Responses to Questions

**Comments to the Author**

1. Is the manuscript technically sound, and do the data support the conclusions?

Reviewer #1: Yes

2. Has the statistical analysis been performed appropriately and rigorously? 

Reviewer #1: Yes

3. Have the authors made all data underlying the findings in their manuscript fully available?

Reviewer #1: Yes

4. Is the manuscript presented in an intelligible fashion and written in standard English?

Reviewer #1: Yes

5. Review Comments to the Author

Reviewer #1: This is an interesting study and authors demonstrated how HGFR inhibition modulates the effect of tumor derived self-DNA on TLR9 signaling and autophagy response by examining the metabolic activity and proliferation of HT29 colon cancer cells that can be considered as continuation of authors previous study entitled, “Modified Genomic Self-DNA Influences In 608 Vitro Survival of HT29 Tumor Cells via TLR9- and Autophagy Signaling” (Ref. 21). The authors have collected a unique dataset using cutting edge methodology. The paper is generally well structured.

However, in my opinion author can proof the mechanism in another colon cancer cell line with an inducible or overexpressed system beside HT29 cells with constitutive expression of TLR9.

The authors should execute more molecular biology based experiments such as rtPCR or western blot of most important genes or proteins respectively to show the signaling pathway and metabolic activity based mechanism.

In the methodology part, some lines are exactly imitated from Ref.21. The authors should take care to avoid plagiarism, specially “HT-29 cell treatment”, “C-MET, TLR9 and autophagy immunocytochemistry”, and “Transmission electron microscopy for evaluation of autophagy” portion.

The authors may also include the following reference on Autophagy in the introduction section along with 13,

i) A novel betulinic acid analogue ascertains an antagonistic mechanism between autophagy and proteosomal degradation pathway in human colon carcinoma cells (HT29). BMC Cancer, 2016, 16:23, 1-19.(DOI: https://doi.org/10.1186/s12885-016-2055-1)

The authors may also include the following reference on Aptamer targeted therapy for colon cancer, in the introduction section along with 19,

ii) Aptamer-Conjugated Apigenin Nanoparticles To Target Colorectal Carcinoma: A Promising Safe Alternative of Colorectal Cancer Chemotherapy, ACS Appl. Bio Mater, 2018. (DOI: 10.1021/acsabm.8b00441)

6. PLOS authors have the option to publish the peer review history of their article (what does this mean?). If published, this will include your full peer review and any attached files.

Reviewer #1: No

---

## [Author Response · Author response to Decision Letter 0]

2 Mar 2022

All the Journal Requirements have been fulfilled during the revision process.

---

## [Decision Letter · Decision Letter 1]

26 Apr 2022

Survival of HT29 cancer cells is influenced by hepatocyte growth factor receptor inhibition through modulation of self-DNA-triggered TLR9-dependent autophagy response

PONE-D-21-29031R1

Dear Dr. Sipos,

We’re pleased to inform you that your manuscript has been judged scientifically suitable for publication and will be formally accepted for publication once it meets all outstanding technical requirements.

Kind regards,

Vladimir Trajkovic

Academic Editor

PLOS ONE

Additional Editor Comments (optional):

Reviewers' comments:

Reviewer's Responses to Questions

**Comments to the Author**

1. If the authors have adequately addressed your comments raised in a previous round of review and you feel that this manuscript is now acceptable for publication, you may indicate that here to bypass the “Comments to the Author” section, enter your conflict of interest statement in the “Confidential to Editor” section, and submit your "Accept" recommendation.

Reviewer #1: All comments have been addressed

2. Is the manuscript technically sound, and do the data support the conclusions?

Reviewer #1: Yes

3. Has the statistical analysis been performed appropriately and rigorously? 

Reviewer #1: Yes

4. Have the authors made all data underlying the findings in their manuscript fully available?

Reviewer #1: Yes

5. Is the manuscript presented in an intelligible fashion and written in standard English?

Reviewer #1: Yes

6. Review Comments to the Author

Reviewer #1: (No Response)

7. PLOS authors have the option to publish the peer review history of their article (what does this mean?). If published, this will include your full peer review and any attached files.

Reviewer #1: No

---

## [Editor Report · Acceptance letter]

4 May 2022

PONE-D-21-29031R1 

Survival of HT29 cancer cells is influenced by hepatocyte growth factor receptor inhibition through modulation of self-DNA-triggered TLR9-dependent autophagy response 

Dear Dr. Sipos:

I'm pleased to inform you that your manuscript has been deemed suitable for publication in PLOS ONE. Congratulations! Your manuscript is now with our production department. 

Kind regards, 

on behalf of

Prof. Vladimir Trajkovic 

Academic Editor

PLOS ONE